# Immunotherapy for the Treatment of Squamous Cell Carcinoma: Potential Benefits and Challenges

**DOI:** 10.3390/ijms23158530

**Published:** 2022-08-01

**Authors:** Tuba M. Ansary, MD Razib Hossain, Mayumi Komine, Mamitaro Ohtsuki

**Affiliations:** Department of Dermatology, Faculty of Medicine, Jichi Medical University, Shimotsuke 329-0498, Tochigi, Japan; tuba2020@jichi.ac.jp (T.M.A.); razib@jichi.ac.jp (M.R.H.); mamitaro@jichi.ac.jp (M.O.)

**Keywords:** nonmelanoma skin cancers (NMSCs), squamous cell carcinoma (SCC), risk factors of SCC, immunotherapy

## Abstract

Melanoma and nonmelanoma skin cancers (NMSCs) are recognized as among the most common neoplasms, mostly in white people, with an increasing incidence rate. Among the NMSCs, squamous cell carcinoma (SCC) is the most prevalent malignancy known to affect people with a fair complexion who are exposed to extreme ultraviolet radiation (UVR), have a hereditary predisposition, or are immunosuppressed. There are several extrinsic and intrinsic determinants that contribute to the pathophysiology of the SCC. The therapeutic modalities depend on the SCC stages, from actinic keratosis to late-stage multiple metastases. Standard treatments include surgical excision, radiotherapy, and chemotherapy. As SCC represents a favorable tumor microenvironment with high tumor mutational burden, infiltration of immune cells, and expression of immune checkpoints, the SCC tumors are highly responsive to immunotherapies. Until now, there are three checkpoint inhibitors, cemiplimab, pembrolizumab, and nivolumab, that are approved for the treatment of advanced, recurrent, or metastatic SCC patients in the United States. Immunotherapy possesses significant therapeutic benefits for patients with metastatic or locally advanced tumors not eligible for surgery or radiotherapy to avoid the potential toxicity caused by the chemotherapies. Despite the high tolerability and efficiency, the existence of some challenges has been revealed such as, resistance to immunotherapy, less availability of the biomarkers, and difficulty in appropriate patient selection. This review aims to accumulate evidence regarding the genetic alterations related to SCC, the factors that contribute to the potential benefits of immunotherapy, and the challenges to follow this treatment regime.

## 1. Introduction

The most frequent cancers in white populations are melanoma and nonmelanoma skin cancers (NMSCs), which account for almost 20 to 30% [1,2]. Although melanoma causes the majority of skin cancer deaths, it is only responsible for 1% of all skin cancers in the United States [3]. The incidence of NMSC cases is continuously increasing because of better cancer screening processes, increases in the elderly population, and increased exposure to ultraviolet radiation (UVR) as a consequence of ozone-layer depletion [4,5]. According to a report published in 2012, which estimates the NMSC incidence in US people, approximately 5.4 million NMSC or keratinocyte carcinoma cases were diagnosed; however, the exact epidemiology of NMSC is unknown because, in the USA, NMSC is not included in the cancer registries [1,6]. The major types of NMSCs are basal cell carcinoma (BCC) and the squamous cell carcinoma (SCC), which account for almost 99% of NMSC; the other types of NMSC include Merkel cell carcinoma, sebaceous gland carcinoma, apocrine adenocarcinoma, angiosarcoma, and dermatofibrosarcoma protuberans [7,8]. BCC is the most prevalent type among all skin cancers in all ethnic groups, although the BCC-to-SCC ratio varies between 1:1 and 10:1 depending on geography, age, and sex [9]. BCC is the most common skin cancer and is more frequently observed in white people compared to SCC, but SCC has the second-highest incidence rates with high metastasis and mortality rates, which is clinically more problematic than BCC [10,11]. The increasing incidence rate of SCC causes a significant financial burden for the patients and one report demonstrated that compared to non-SCC patients the average in-patient cost for the SCC patient is about USD 0.066 million [12]. In general, the prognosis of SCC is quite good with a 5-year survival rate of over 90% [4]. According to a report published by the *Australasian Journal of Dermatology* and the American Cancer Society, from 200 to 2000 deaths have been reported that were due to the NMSC, mostly SCC, in Australia and the USA, respectively [13,14]. The most significant risk factors for SCC include UVR exposure, radiotherapy, fair complexion, and age [15,16]. Primarily, for light-toned people, it develops from DNA mutations caused by prolonged exposure to UVR, mainly in the squamous cells, and it can metastasize through blood and lymph vessels, and cause serious complications [17]. The major treatment modalities for SCC include surgical excision, which can cure 95% of cases if diagnosed early. In the advanced stages when resection is difficult, other treatment options are available, such as, cryotherapy and curettage, radiotherapy, Mohs microscopic surgery, and topical medication [18,19]. SCC is characterized by massive infiltration of inflammatory cells, high expression of immune co-signaling molecules, immunosuppression, and high mutational load, and in 2018, the U.S. Food and Drug Administration (FDA) approved cemiplimab-rwlc (Libtayo), an anti-PD-1 (Programmed Death 1) antibody, as the first immunotherapy to treat patients with metastatic cutaneous squamous cell carcinoma (CSCC) [20,21]. In addition, pembrolizumab and nivolumab are approved for the treatment of advanced SCC, which showed better survival in patients with esophageal SCC and head and neck SCC in Japan, respectively [22]. The success of immunotherapy for treating SCC patients is challenging because of the SCC patients’ advanced age, immune resistance, comorbidities, and possible adverse side effects of immunotherapies. Our objective in this review is to elucidate the possible benefits and challenges of immunotherapies to treat SCC patients.

## 2. The Etiology of SCC

SCCs are among the most frequent human malignancies and represent almost 20% of all skin-cancer-related deaths characterized by accelerated and abnormal growth of squamous cells. Squamous cells are one of the epithelial cells that can be found in the skin epithelium and the mucous membrane [23]. SCCs can develop in different anatomical sites, such as, skin, esophagus, oral and nasal cavity, salivary glands, lung, genitals, and urinary tract, and, although rare, they are extremely fatal when they develop in the thyroid, prostate, scalp, and breast [24,25,26,27]. In the white population, almost 80% of SCCs mainly develop in chronic sun-exposed areas; specifically, almost half a percentage of SCC cases in Caucasians occur in the hands, head, and neck, which implies that UVR is the most important carcinogen for SCC [28,29]. The other non-sunlight-related extrinsic risk factors that contribute to SCC include, exposure to radiotherapy and chemical substances, habitual factors, 8-methoxypsoralen (P), and ultra-violet light (UVA) (PUVA) treatment, and medications. The intrinsic risk factors involved in SCC development include older age, skin pigmentation, history of immune suppression, history of actinic keratosis (AK), chronic medical conditions, viral infections, chronic wounds, hereditary conditions, and personal history of NMSC [17,30,31,32,33,34,35,36,37,38,39,40,41,42] (Table 1).

SCC is commonly known as one of the human cancers that result from the increased accumulation of genetic mutations caused by various carcinogens. Recent studies have demonstrated the commonly mutated genes from different pathways known as cell cycle regulators, keratinocyte differentiation, NOTCH-signaling pathways, tumor suppressor pathways, mitogenic/survival-signaling pathways, cyclooxygenase pathways, Hippo-signaling pathways, etc. (Figure 1).

### 2.1. External-Factors-Associated Genetic Mutation

UV exposure is one of the most important environmental contributing factors to NMSC, as almost 90% of NMSC cases are associated with UVR [43]. UV radiation causes cellular DNA mutation resulting in accelerated cell growth, inactivating tumor-suppressor genes, and formation of the tumor [43]. Based on the wavelength, UVR can be classified into three types: UVA (320–400 nm), UVB (280–320 nm), and UVC (100–280 nm) [44]. Among them, almost all UVC and about 90% UVB are absorbed by the ozone layer. UVC cannot have any impact on our skin [45,46].

The rest of the UVB can reach the Earth’s surface, irradiate the skin epidermal keratinocytes, and cause direct DNA damage; while UVA has the longest wavelength and it causes indirect DNA damage through oxidative stress pathways and by forming 8-oxoguanine, which results in GC→TA transversion mutations during replication [44,47,48]. The two major types of photoproducts are cyclobutane pyrimidine dimers and photoproducts produced by UVB and UVC irradiation and they are responsible for photolesions on DNA, DNA mutation, and skin cancer [49]. The most commonly mutated gene in SCC is *TP53*, which occurs in almost 50–90% of cases and is induced by UVR [50,51]. One report demonstrated that *p53-deficient ^–/–^* mice are more likely to develop SCC after UV irradiation [52]. *TP53* encodes tumor-suppressor protein *p53*, plays important functions as a transcription factor, and under stress conditions it regulates cell cycle, apoptosis, senescence, and DNA repair [4]. In response to a DNA damage incident such as, UVR or radiation, the *p53*- *p21*^Cip1^ pathway gets activated, inhibiting cyclin-dependent kinase (CDK), which initiates cell cycle arrest in the G1-S phase to allow the DNA damage response [53]. Thus, *p53* is known as a guardian of the genome in some cases [54]. In other cases, when the DNA damage response is not initiated, the damaged cells are eliminated by the *p53*-mediated apoptosis [55]. In human sun-exposed skin areas, various photoproducts are activated by cyclobutane pyrimidine dimers (CPDs), and pyrimidine-pyrimidone causes C-T or CC-TT mutation in the *p53* gene, about half of these mutated cells die by apoptosis and other mutated cells expand, increase in size, and develop carcinogenesis [56]. Usually, in most individuals, the event of sunlight exposure and *p53* mutation can occur early in life, which may lead to precancerous lesions, actinic keratosis, and finally develop into SCC in later life [50]. The cyclin-dependent kinase inhibitor 2A (CDKN2A) locus located on human chromosome 9 *p21* can be found mutated in SCC because of point mutation, promoter hypermethylation, and loss of heterozygosity [57]. CDKN2A encodes two tumor suppressor genes, *p16INK4a (p16)* and *p14ARF (p14)*, and both of them perform crucial cell-cycle regulatory functions. The tumor suppressor gene *p16* is a negative regulator of cell proliferation that inhibits CDK4, prevents retinoblastoma phosphorylation (pRB), blocks E2F-mediated transcription, and sequentially induces G1-S phase cell cycle arrest and senescence [58]. Similarly, *p14* also has tumor-suppressor effects that antagonize or neutralize the MDM2, which is a negative regulator of *p53*, and by neutralizing MDM2, *p14* allows the *p53* transcriptional activity. Therefore, the *p14* and *p16* gene mutations or loss of function can lead to uncontrolled cell proliferation, a common feature observed in carcinogenesis [58]. It has been reported that the *p16* mutation is observed in almost 24% of cases of SCC patients [59]. There is a positive correlation between *p16* overexpression and cancer progression from actinic keratosis to invasive SCC and *p16* overexpression is mostly located in UV-exposed skin tumors [60,61]. UVB irradiation plays an important contributing factor for *p16* mutation as proven by the previous reports where *p16* mutation is mostly observed in actinic keratosis, which is a common precancer lesion caused by sun exposure [61]. However, there is no clear indication of how UV irradiation causes the *p16* mutation. One report speculated that the *p16* mutation found in SCC can be caused by CPD photoproduct from UVB [59]. Another report examined tumors from 40 SCC patients and showed UVR-signature-type mutations in dipyrimidine sites of human skin, which are positively correlated with the *p16* and *p14* mutations and that proves the association between the mutagenesis in these two tumor-suppressor genes and UV-induced SCC [58]. Another report showed the reduced expression of tumor suppressor protein pRB by immunohistochemical analysis in about 8% of actinic keratosis patients and 16% of SCC patients [62].

The telomerase reverse transcriptase (TERT) mutation is regarded as a comprehensive feature observed in widespread malignancies and represents a UV-irradiation signature type [63]. One report demonstrated frequent TERT mutation in SCC (50%) in which all of them are C > T or CC > TT types observed in dipyrimidine sites characterizing the UVR-related mutagenesis [64].

Arsenic is one of the most poisonous metalloids and also a well-known human carcinogenic factor found ubiquitously in nature. It is documented that arsenic might not have a direct carcinogenic effect but it works as a co-carcinogen with UV and contributes to melanoma and NMSC malignancies by enhancing the genotoxicity of UVR [65]. Bowen’s disease is one of the precancerous lesions (SCC in situ) possibly caused by chronic arsenic exposure and has a chance to transform into a more aggressive phenotype [66]. The precise mechanism of arsenic-induced carcinogenesis is still unknown. One report compared the production of reactive oxygen species (ROS) in arsenic-related and non-related skin neoplasms and found that ROS, specifically 8-hydroxy-2′-deoxyguanosine, showed higher expression in 78% of arsenic-related skin cancers [67]. Another report demonstrated the presence of *p53* and *p16* hypermethylation in the blood samples from subjects who have a history of arsenic-contaminated water ingestion and symptoms of skin cancer [68]. Another possible mechanism may be involved in histone methylation dysregulations or cyclin D promoter unmethylation caused by arsenic toxicity [69]. Hexavalent chromium is another possible carcinogen found in the environment. One murine study showed that potassium chromate in drinking water along with UV irradiation cause skin tumorigenesis in a dose-dependent manner, which may be involved in ROS-dependent pathways [70].

One meta-analysis and systemic review showed a strong association between smoking and increased risk of cutaneous SCC [71]. Among the habitual factors, cigarette smoking and alcohol consumption are strongly associated with head and neck SCC (HNSCC) and whole-exome sequencing analysis demonstrated that among 92 HNSCC patients, 89% have a smoking history and 79% have an alcohol consumption history [72]. Another whole genome sequencing analysis using 32 primary tumors confirmed smoking is associated with more genetic mutations compared to non-smokers (21.6:9.5) [73]. Tobacco is associated with many epigenetic alterations; for example, it causes overexpression or mutation of *TP53*, *p16*, *TP63*, *NOTCH1*, and *PI3K* (the phosphatidylinositol 3-kinase), oxidative stress, inhibiting immune functions, and is involved in the oral squamous cell carcinoma (OSCC) pathogenesis [72,74]. PUVA, a widely used photochemotherapy for psoriasis, is also associated with an increased incidence rate of cutaneous SCCs [75].

The widely known treatment for stage IV metastatic melanoma is BRAF (B type rapidly accelerated fibrosarcoma kinase) inhibitors and with this monotherapy, NMSCs are the most commonly observed skin tumors. There are two possible hypotheses available in the literature to uncover the molecular mechanism. The first theory suggested that in the case of BRAF-mutated skin cancers, BRAF inhibitors inhibit MAPK (mitogen-activated protein kinase)-signaling pathways (pERK), causing cell cycle arrest, apoptosis, and inhibiting tumor progression; however, in the case of wild-type BRAF cells, the ERK is activated and causes abnormal cell proliferation, which may lead to SCC development [76,77]. According to the second theory, the BRAF inhibitor causes keratinocyte hyperproliferation by inducing CRAF, which leads to MAPK activation and results in SCC formation [78,79]. In advanced BCC, inhibition of the hedgehog pathway by vismodegib has recently been approved by the FDA [79]. Although SCC development has been observed during the hedgehog-pathway-inhibitor treatment, which may include the activation of the *RAS/MAPK* pathway, there are some controversial reports regarding hedgehog-pathway-inhibitor-induced SCC formation [80]. The FDA approved JAK (Janus kinase) inhibitor ruxolitinib for the treatment of myelofibrosis and polycythemia vera, which can be associated with potential SCC development. A 10-year clinical trial was conducted on 564 patients and showed an increased risk of developing SCC after ruxolitinib treatment [81]. However, the mechanism of how ruxolitinib increases the risk of SCC development is as yet unclear. It is speculated that the immune function gets inhibited by the JAK inhibitors, specifically, the production of IL-6, IL-23, and Th17 activity, which may favor tumor formation during SCC development [82]. A meta-analysis showed a positive correlation between antihypertensive drugs, specifically thiazide diuretics, and the risk of SCC development. That report also suggested that thiazide diuretics can increase UVR-induced photosensitivity by stimulating photoproducts [83]. Voriconazole is an approved antifungal medication for transplant patients that modifies the forkhead box-1 *(FOXM1)* and cyclooxygenase-2 *(COX-2)* expression, which is commonly found upregulated in SCC patients [84].

### 2.2. Internal-Factors-Associated Genetic Mutation

Due to the increased risk of recurrence and metastasis, patients with immunosuppression are at a high risk of SCC-associated deaths, and in the case of organ transplant recipients (OTRs), SCC is the most frequently observed post-transplant cancer. The OTRs have 65–250-fold higher SCC incidence rates compared to others and these recipients are at 5–8% more risk of developing metastatic SCC [85]. Immunosuppressants are normally prescribed for OTRs to prevent organ rejection by targeting the immune system, which thus provides a favorable environment for carcinomas. The molecular pathways related to the immunosuppressive agents and SCC progression might include the classic, p53, NOTCH1 pathway, or other inflammatory pathways. For example, azathioprine, an immunosuppressive-agent-treated OTR, is susceptible to developing SCC and has a mutation in TP53, CDKN2A, or NOTCH1/2/4 [57]. The inflammatory cytokines and the immune cell infiltration are also associated with SCC development. It has been reported that the calcineurin-inhibitor drugs, specifically ciclosporin, caused an upregulation of interleukin-22 expression that has been reported to increase SCC proliferation and migration [86].

Sufficient evidence has proven the positive association between chronic wound and cutaneous SCC formation. Matrix metalloproteinases (MMPs) play an important role in extracellular metrics and basement membrane degradation and thus fuel tumor metastasis. In a study, expression of MMP-7, MMP-12, and MMP-13 in venous leg ulcer patients have a positive correlation with malignant progression of SCC [87].

Among genodermatosis, patients with the epidermolysis bullosa (EB), characterized by blistering and fragile skin and mucosa, are 70 times more susceptible to being diagnosed with SCC compared with the general population [88]. The EB-associated SCC is highly aggressive and associated with high mortality. Immune deficiency, loss of collagens, and mutations in TP53 and NOTCH family members in EB cause a favorable microenvironment for SCC development [88,89].

## 3. Available Treatments for SCC

The SCC represents a multi-stage progression process from actinic keratosis (SCC in situ) and precancerous lesions to locally advanced SCC, and in very rare cases (about 3–9%) to metastatic SCC [90]. Because of this spectrum of conditions, the therapeutic approaches also vary according to the severity of the stages of SCC. The treatment options for actinic keratosis or precancerous lesion include two broad categories; lesion-directed therapy and field-directed therapy. The lesion-directed therapies targeted to remove the specific lesions include 1. cryosurgery (use of liquid nitrogen), 2. curettage, 3. dermabrasion, 4. photodynamic therapy using aminolaevulinic acid or methylaminolaevulinic acid, and 5. laser therapy. On the other hand, the field-based therapies target multiple lesions to reduce the burden and this approach includes topical application of 5-fluorouracil (5-FU), imiquimod, tirbanibulin, diclofenac sodium, and chemical peels [91]. The most common treatment modalities available for locally advanced SCCs and metastatic SCCs are excisional surgery, or Mohs surgery. To completely remove the SCC tumor with minimum excision, the Mohs microscopic surgery is considered the gold standard because it showed a lesser recurrence rate as it can remove 100% SCC cell margins and can be applied to tumors that are recurrent, large, or have distinctive edges [92]. In some cases, the patients’ specific conditions, for example, age or co-morbidities, or tumor locations do not allow or make it difficult to perform surgical excision. On that occasion, physicians suggest performing radiation therapy, sometimes after the surgery or in combination with other treatment modalities, such as chemotherapy or EGFRis (epidermal growth factor receptor inhibitors). There are a wide range of radiation therapies available, such as, superficial or orthovoltage x-rays, use of electron beams, and high-dose-rate brachytherapy [93]. There are limited clinical data regarding the systemic treatment with chemotherapy in the treatment regimen of SCC. Nowadays, the most commonly used inductive chemotherapy includes a combination of 5-FU (fluorouracil), cisplatin, and docetaxel, which is associated with better survival with less toxicity. The 5-FU can be used as a monotherapy and one paper showed oral 5-FU treatment displayed 64.3% overall improvement in 14 elderly patients [94]. However, the toxicity, cost, and short-duration nature are still major concerns related to chemotherapy. In targeted therapy, a specific drug is used to target a molecule or factor that is responsible for the cancer cell growth or development. In SCC, it has been reported that 80% of SCC tumors and almost all of the metastatic tumors expressed high EGFR, which is positively correlated with SCC progression and morbidity [95]. The well-known and FDA-approved EGFR-targeted therapy includes cetuximab, gefitinib, and osimertinib for HNSCC [96,97]. Patients with advanced SCC demonstrated 16% and 53% overall response rate (ORR) after treating with gefitinib and cetuximab, respectively [98,99].

In immunotherapy, medicines are used to strengthen a patient’s immune system, which can identify and recognize the cancer cells from the normal cells and can destroy only the cancer cells more efficiently. Immunotherapies, specifically the immune checkpoint therapies, are the novel drugs initially developed to treat melanoma, which mainly targets the cytotoxic T-lymphocyte-associated antigen 4 (CTLA-4) and programmed cell death 1 (PD-1). Under normal physiological conditions, the immune systems, specifically the T cells, protect our body from foreign invaders, leaving the normal healthy cells. The PD-1/PD-L1 pathway is recognized as an immune checkpoint to halt or turn off the T cell response, resulting in protecting normal cells from T cell attack and also in minimizing inflammatory response. In patients with cancer, the PD-1 receptor is expressed in activated T cells and antigen-presenting cells, and tumor cells express PD-1 ligand and programmed cell death-ligand 1 (PD-L1). The binding of PD-1 with its ligand decreases T cell functionality, suppresses the immune system function, and accelerates cancer cell proliferation. The PD-1 antibody blocks the PD-1 interaction with its ligand PD-L1/PD-L2 and allows the immune system to destroy the tumor cells by increasing cytotoxic cells [92]. Later in 2018, the FDA approved cemiplimab-rwlc (Libtayo), a PD-1 inhibitor, as the first immunotherapy to treat patients with metastatic cutaneous SCC. The Phase I and Phase II studies with cemiplimab showed almost 65% and 61% durable disease control in advanced cutaneous SCC patients, respectively [20]. The other PD-1 antibodies, pembrolizumab (Keytruda^®^, MSD (Merk Sharp and Duhme) & Co., Inc., Rahway, NJ, USA), and nivolumab (OPDIVO, Bristol-Myers Squibb Co., New York, NY, USA), were recommended in 2020 by the FDA to treat patients with recurrent or metastatic HNSCCs and esophageal squamous cell carcinoma (ESCC) after prior fluoropyrimidine- and platinum-based chemotherapy, respectively [100,101] (Table 2). Other checkpoint inhibitors, such as, Erlotinib and Ipilimumab, that might have beneficial effects with advanced SCC patients are under consideration [102,103,104,105].

### 3.1. Factors That Determine the Potential Benefits of Immunotherapy in SCC Treatment

In some cases, tumors characterized as advanced, recurrent, located in cosmetically essential areas, or metastatic are not eligible for surgical excision or radiotherapy. On those occasions, systemic therapies, especially immunotherapies, are the most recommended therapies among other treatments. Cemiplimab, a highly potent human monoclonal PD-1 antibody, is efficient and safe to treat advanced SCC, as shown by phase I and II clinical trials. The objective response rate was about 50% and 47% in Phases I and II, respectively [106,107]. The characteristics of the tumor microenvironment (TME) of SCC and higher SCC prevalence among immunocompromised patients provide a strong rationale for the efficacy of immunotherapy in SCC management (Figure 2).

Understanding of the TME landscape and the immune regulation is absolutely essential for the development and better outcome of therapies. The TME in SCC is composed of a stromal component with cancer-associated epithelial, endothelial cells and fibroblasts, melanocytes, vascular cells, infiltrating immune cells ((T cells, B cells, natural killer cells, dendritic cells, tumor-associated macrophages (TAMs) myeloid-derived suppressor cells, and tumor-infiltrating lymphocytes (TILs); CD4, CD8, regulatory T cell), and non-cell components of extracellular matrix composed of proteins; fibronectin, collagen, glycoprotein, MMPs, laminins, tenascin, gelatin and syndecan, vitronectin, growth factors (EGF, fibroblast growth factor (FGF, hepatocyte growth factor, vascular endothelial growth factor (VEGF)), cytokines (TNF- α; tumor necrosis factor-α, interleukins (ILs) and chemokines; C-X-C motif chemokine ligands (CXCLs; CXCL,1,8,12,14 and C-C motif chemokine ligands (CCLs; CCL2,5,7) [108,109,110].

Cumulative evidence suggests that several TME factors, for example, TILs (CD4, CD8), growth factors, Tregs, and checkpoints (PD-1, CTLA-4), are correlated with the efficacy of checkpoint inhibitors in SCC. The TIL represents the immune therapy response to tumor cells and thus the density of TILs can predict immune checkpoint inhibitors’ efficiency and is recommended as a favorable prognostic biomarker. Among the TILs, CD3, CD8 (effector/cytotoxic), and CD4 (helper) T cells are recognized to have an anti-tumor immune response and found to positively correlate with SCC patients’ favorable outcomes. One report showed that among 63 patients with HNSCC, all of the tumors have TILs positive for CD3 and CD4 immunohistochemistry, and increased CD4-positive cell infiltration was associated with improved survival compared with the poor CD4-positive cell-infiltrating tumors [111]. In another report, 50 patients with OSCC were assessed for CD8- and CD4-positive cells and they found that CD8-positive cells were more abundant than CD4 in tumor specimens and there is a negative correlation between CD8 infiltration and metastasis [112]. However, another study used multicolor flow-cytometry analysis in SCC lesions and found comparatively lower CD8 T cell infiltration. They also showed that a transforming growth factor β (TGF-β)-overexpressing SCC cell line decreased CD8-positive cell infiltration, showing that TGF-β might be responsible for decreasing dendritic cell and CD8-positive cell infiltration [113,114]. The efficacy of PD-1 inhibitor is associated with CD8-positive T cell infiltration as suggested by a report that showed CD8-positive cytotoxic T cell depletion caused a complete abolition of the PD-1 inhibitor’s efficiency and increased tumor growth [115].

Another factor recognized as an independent prognostic factor in various cancers for immunotherapy is tumor mutational burden (TMB). Cutaneous SCC is associated with high TMB; more precisely, the whole exome analysis revealed almost 50 mutations/Mbp DNA [57]. It has been shown that tumors with high TMB analyzed by hybrid capture-based next-generation sequencing (NGS) showed a better outcome in patients treated with PD-1/PD-L1 monotherapy [116]. However, in some cases, the predictive value of TMB is less. In the case of recurrent and metastatic HNSCC, it has been recommended to use a combination of several gene mutations (PIK3CA, TP53, or ROS1 mutation) and TMB value (known as TP-PR prognostic index) for better prediction of the outcome of immunotherapy. They also reported that patients with high TP-PR showed increased CD4^+^ and CD8^+^ cells and better overall survival undergoing immunotherapy compared with those with low TP-PR scores [117]. It has been reported by an NGS analysis that UVR is associated with diverse genomic mutation related to high TMB, and that among these UVR signatures the commonly mutated genes are *TP53*, *NOTCH1/2/3*, *TERT*, *CDKN2A*, and *RB1,* which are also associated with SCC [118].

In most solid tumors, the tumor-associated macrophages (TAMs) are associated with bad prognostic factors as they inhibit CD8^+^ cells, and secrete cytokines and growth factors (FGF, VEGF, EGF, MMPs, and TGF-β) favorable for tumor growth and metastasis [119]. It has been reported by immunohistochemical analysis in oral SCC-tumor specimens that TAMs are the major group of leukocytes in the SCC TME and CD163^+^/CD204^+^ M2 macrophages are the major TAMs recruited in the oral SCC TME [120]. Another report showed that in the TME of the cutaneous SCC, CD163 macrophages are the most abundant among other TAMs, which is associated with increased MMP (MMP9 and MMP11) expression. MMPs are thought to promote SCC aggressiveness and malignancy by degrading extracellular matrix and promoting angiogenesis [121,122].

Another factor that determines the efficiency of the checkpoint inhibitors is the expression of PD-L1 in the tumor cells. Generally, increased PD-L1 expression is associated with better prognosis defined as a better response rate and overall survival. An immunohistochemical analysis demonstrated that PD-L1 expression was higher in tumor cells of SCC (52%) compared to that in adenocarcinoma. Moreover, the PD-L1 expression in the tumor-infiltrating immune cells was found to be associated with the tumor stage and survival rate, showing that almost 40% of tumor-infiltrating immune cells of the advanced esophageal SCC patients showed positive expression of PD-L1 [123]. However, the use of PD-L1 as a predictive biomarker is still controversial. Expression of PD-L1 as a poor or favorable prognostic marker depends on where it expressed, in the tumor cells or in the tumor-infiltrating immune cells [123].

SCC is characterized by increased genetic mutations and chromosomal instability. Generally, there is a positive correlation between genetic instability and immunotherapy effectiveness as genomic instability causes better T cell recognition and tumor death. Accumulation of gene mutations and increased genetic mutations in the SCC tumor microenvironment positively favor response to immunotherapy. It has been reported that activation of Wnt- and TGF-β-signaling pathways attenuate the effectiveness of immunotherapies and also causes resistance to immunotherapies [124]. One report showed that increased TGF-β1 expression in human oral SCC (OSCC) cells is associated with reduced function of cytotoxic T-lymphocytes and thereby produces an immunosuppressive tumor microenvironment [125]. Another report confirmed the TGF-β receptor II mutation in OSCC cells, which is associated with tumor progression [126]. Previous reports have confirmed that the aberrant expression of Wnt-signaling molecules is associated with the SCC tumor invasiveness and metastasis and that several Wnt molecules, for example, Dickkopf 3 (DKK3), secreted frizzled-related protein (SFRP) and the Wnt inhibitory factor (WIF) can be used as biomarkers [127]. Moreover, it is well-known that Wnt-signaling activation is associated with immunotherapy resistance, although more research is needed to confirm the association between Wnt signaling and immunotherapy resistance in SCC treatment.

Based on the characteristics described above, SCC is highly immunogenic, which implies the possible reasons for positive outcomes of immunotherapy.

### 3.2. Potential Challenges of Immunotherapy in SCC Treatment

Although immunotherapies have appeared as a major treatment modality for cutaneous SCC patients, there are some challenges that medical professionals need to deal with (Figure 3).

The first challenge is to select the appropriate patients best fit for the immunotherapy to achieve the best clinical outcomes. It has been reported that patients with a high tumor mutational burden and diagnosed with advanced cutaneous SCC have more clinical benefits from immunotherapy [128]. This finding is in alignment with the clinical trials using cemiplimab, which showed a 65% and 61% durable disease control in patients with advanced cutaneous SCC and metastasis in Phase I and Phase II, respectively [20]. There is mixed information about using the PD-L1 status of the SCC tumor as a prognostic marker. In the clinical study using nivolumab, it has been shown that the improvement of overall survival observed in the nivolumab group was not associated with the PD-L1 expression [90]. Other studies have also shown no association between PD-L1 expression and immunotherapy efficiency [129,130]. In another study, the objective response rate was increased from 4% to 22% when patients had PD-L1 expression in both tumor cells and tumor-infiltrating immune cells. On the other hand, increased PD-L1 was associated with a reduced survival rate in HNSCC [131]. In this regard, the FDA had approved a PD-L1 immunohistochemistry kit (Dako PD-L1 IHC 22C3 pharmDx) to choose patients appropriate for pembrolizumab monotherapy [132]. Another biomarker to choose responders for immunotherapy is the immune gene expression profile scores, which analyze the T cell activation condition in the TME. In one study, tumors from patients with different cancers including HNSCC were analyzed before pembrolizumab or nivolumab treatment to identify the genes associated with a better outcome. They found that tumors with high mRNA expression of cytotoxic T cells, natural killer cells, CD8^+^, CD4^+^, Treg, interferon-gamma (IFN-γ), PD-L1, and PD-1 had more non-progressive disease and progression-free survival [133,134]. Therefore, it can be suggested that patients with an increased T cell adaptive immune system at the baseline level have more susceptibility to benefit from immunotherapy.

Another challenging factor for immunotherapy is the primary or acquired resistance that occurred during the treatment and because of this reason the number of patients who positively respond to this monotherapy is relatively low. The process of this resistance is really complex and encompasses a broad spectrum of mechanisms with individual differences. IFN-γ, a key cytokine in the innate and adaptive immune system in the TME produced by the activated T cells or natural killer cells, and IFN-γ can cause resistance toward immunotherapy through PL-L1-dependent or -independent pathways. IFN-γ was found to increase the PD-L1 expression that blunts the inhibition of PD-L1/PD-1 interaction, which is the mainstay of antitumor effects of checkpoint inhibitors, and contributes to the resistance to immunotherapy [135]. Moreover, it has been reported that interferon-induced RNA-editing enzyme, adenosine deaminase acting on RNA 1 (ADAR1), is associated with PD-1 blockage resistance and reports showed deletion of ADAR-1 increased CD8^+^ cell, IFN-related gene expression and thereby improved sensitivity toward the PD-1 antibody [136]. SCC tumors showed increased ADAR1 expression [137,138] and this might contribute to the immune checkpoint inhibitor resistance observed in the SCC treatment. IFN-γ can also contribute to the immunotherapy resistance by phase-separation of yes-associated protein (YAP) in the tumor cells. TME in the HNSCC showed increased YAP expression compared to the tumor-adjacent area, which is associated with PD-L1 transcription and promotes resistance [139,140].

The immune-related adverse events (irAEs) are another challenge that is sometimes observed in the immunotherapy-treated patients and the occurrences of these irAEs are often dependent on the extrinsic factors, for example, type and the dose of the therapy, duration of treatment, or in the intrinsic factors such as prior auto-immune diseases, or other hereditary conditions, patients’ immune system in the TME, etc. [141]. Diarrhea, fatigue, nausea, pruritus, rash, reduced appetite, constipation, and asthenia are the most commonly observed adverse effects seen in the patients treated with FDA approved PD-antibodies cemiplimab, nivolumab, and pembrolizumab.

## 4. Conclusions

Because of the poor outcome and limited treatment options, SCC remains a challenging disease for medical professionals. Among other standard therapies, for example, surgical excision, radiotherapy, and chemotherapy, immunotherapies have evolved as a most promising treatment regime as they are highly tolerable and have fewer adverse effects for the SCC, a highly immunogenic skin cancer. Although the number of patients getting benefits out of immunotherapy is still less because of the increased risk of developing resistance and less availability of reliable biomarkers to choose appropriate patients. Preventive strategies mainly focus on avoiding risk factors by adopting some environmental and lifestyle changes, for example, using sunblock, performing regular skin screening procedure, and intake of chemopreventive agents such as Ellagic acid, black tea, curcumin, phenethyl isothiocyanate, etc. Additional advanced studies are needed to overcome the challenges and to propose a combination of treatment modalities so that a higher number of patients can be cured.

## Figures and Tables

**Figure 1 ijms-23-08530-f001:**
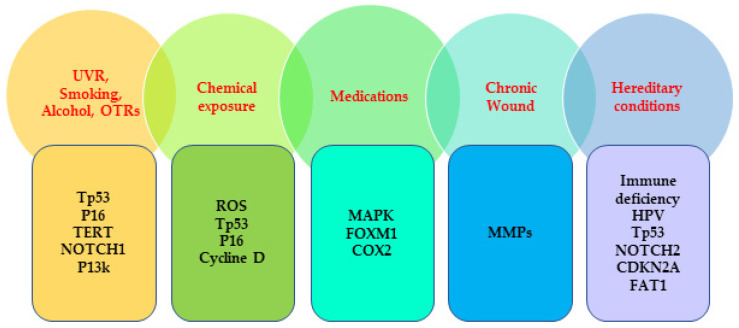
Commonly mutated genes associated with the risk factors of SCC. UVR, ultraviolet radiation; OTRs, organ transplant recipients; TERT, telomerase reverse transcriptase; ROS, reactive oxygen species; MAPK; mitogen-activated protein kinase; FOXM1; forkhead box M1; COX2, Cyclooxygenase 2; MMPs; matrix metalloproteinases, HPV, human papillomavirus; CDKN2A; cyclin-dependent kinase inhibitor 2A; FAT1, FAT atypical cadherin 1.

**Figure 2 ijms-23-08530-f002:**
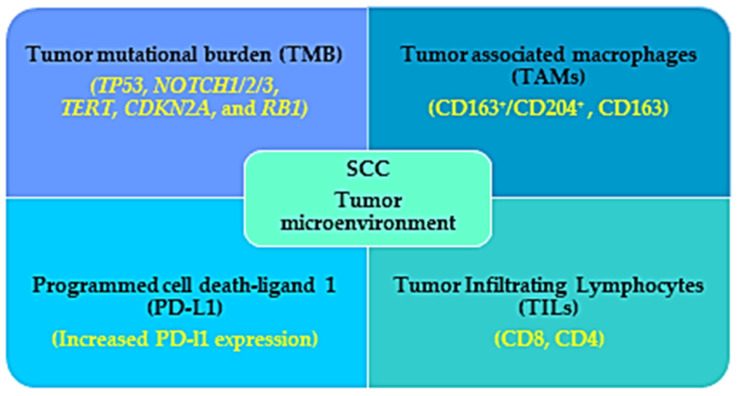
Immunogenic characteristics of SCC that favor the benefits observed in immunotherapy treatment for SCC patients. The tumor microenvironment (TME) of the SCC patients is characterized by high tumor mutational burden (TMB), increased infiltration with tumor associated macrophages (TAMs) and lymphocytes, and increased PD-L1 expression, which work as a favorable prognostic factor for immunotherapy in SCC patients.

**Figure 3 ijms-23-08530-f003:**
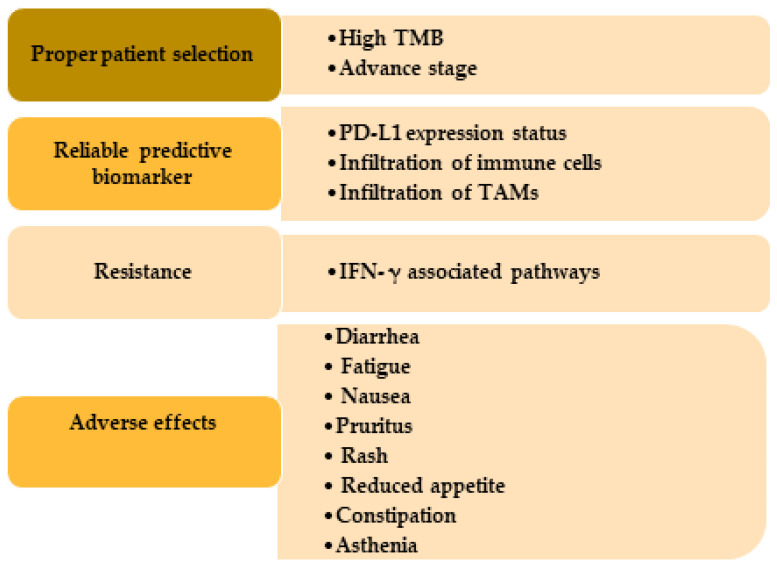
Challenges usually occur in the immunotherapy treatment in SCC patients. Patients with high TMB and advanced or metastatic stage get more benefit from immunotherapy. Expression of PD-L1, immune cells, and TAMs are useful as biomarkers. IFN-γ causes immunotherapy resistance through PD-L1-dependent or -independent pathways and by producing adenosine deaminase acting on RNA 1 and yes-associated protein. TMB, tumor mutational burden; PD-L1, programmed cell death-ligand 1; TAMs, tumor-associated macrophages.

**Table 1 ijms-23-08530-t001:** Major Risk factors for Squamous Cell Carcinoma.

Factor Type	Major Factors	Specific Factors
**Extrinsic Factors**	Ultraviolet radiation (UVR)	Ultraviolet (UV)B, UVA
	Radiotherapy	
	Chemical exposure	nickel, arsenic, chromium, hydrocarbon, pesticides, herbicides, insecticides, fungicides, petroleum products, (gasoline and oil), grease, and diesel fumes
	Habitual factors	cigarette smoking, alcohol consumption, dietary factors (iron deficiency, malnutrition, oral hygiene), 8-methoxypsoralen (P) and UVA (PUVA) treatment
	Medications	BRAF inhibitors; vemurafenib, dabrafenib, sonic hedgehog-inhibiting agents; vismodegib; JAK inhibitors; ruxolitinib, PDE-5 inhibitors, antihypertensive drugs; diuretics, antifungal medication; voriconazole
**Intrinsic factors**	Age	more than 60
	Sex	male
	Skin type	pale skin
	Precancerous lesion	actinic keratosis, Bowen’s disease
	History of immune suppression	post transplantation, cancer therapy
	Chronic medical conditions	organ transplant recipients, chronic wounds (chronic osteomyelitis, chronic venous ulcers)
	Viral infections	HIV/AIDS, human papillomavirus, Epstein–Barr virus, John Cunningham virus
	Hereditary conditions	(xeroderma pigmentosum, oculocutaneous albinism, epidermolysis bullosa, dyskeratosis congenita, Huriez syndrome, epidermodysplasia verruciformis, Rothmund–Thomson syndrome, Bloom syndrome, Werner syndrome, GATA2 deficiency, DOCK8 deficiency, Fanconi anemia
	Past history of NMSC	

UVR, ultraviolet radiation; BRAF, B type rapidly accelerated fibrosarcoma kinase; JAK, Janus kinase; PDE-5, phosphodiesterase type 5.

**Table 2 ijms-23-08530-t002:** FDA approved PD-1 antibody for the treatment of SCC.

Antibody	Target	Study Type	Total Number of Patients (n)	Disease/Tumor Type	Overall Response Rate (ORR)	Reference
Cemiplimab	PD-1	Phase-I, open-label, multicenter study	26	advanced cSCC	50%	[106]
		Phase-II, nonrandomized, global,	59		47%	
Pembrolizumab	PD-1	phase II, global, open-label, nonrandomized,	159	locally advanced or recurrent and/or metastatic cSCC	50%	[100]
Nivolumab	PD-1	Phase III, multicenter, randomized (1:1), active-controlled, open-label	419	unresectable advanced, recurrent or metastatic esophageal SCC (ESCC)	19.3%	[101]

## Data Availability

Not applicable.

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
