# Peer review of "Immunotherapy for the Treatment of Squamous Cell Carcinoma: Potential Benefits and Challenges"

_ijms, 2022, doi:10.3390/ijms23158530_

Round 1

Reviewer 1 Report

In this review Ansary et al extensively describe the field of immunotherapy applied to SCC, describing some aspects regarding the genetic alterations related to SCC, the factors that contribute to the potential benefits of immunotherapy, and the challenges to follow this treatment regime. The topic chosen is undoubtedly very interesting and some changes and rephrasing could improve highly the quality of the manuscript. Please the presence of typos is very common along all the manuscript and should be corrected.

-          The abstract should be shortened in a simpler version that helps legibility, in my opinion too much information is included in this one, particularly the introduction could be highly shortened.

-          Introduction:

-          Any specific details (quantifications) about the medical costs and deaths caused by SCCs would be very appreciated, also some numbers about risk factors and % of early diagnosed cases.

-          Some terms should be better employed as caucasian or white population.

-          <in line 55: Primarily it develops from DNA mutations caused by prolonged ex-
posure to ultraviolet radiation (UVR) mainly in the squamous cells and it can metastasize through blood vessels and lymph nodes, and cause serious complications [12]

-          Line 65: Please rephrase the following to increase legibility: Because of the SCC patients’ advanced age, immune resistance and com-modities and possible adverse side effects of immunotherapies the success of immuno-therapy for SCC treatment is challenging.

-          Line 75: please change fetal by fatal. Some typos and English grammar should be revised along the document.

-          Line 205: the mechanism of how Ruxolitinib increases the risk of SCC development is yet “clear”, should be changed by “unclear”.

-          Please include in 2.2 section the positive correlation between EBDR and SCC predisposition.

-          Table 1 and 2 should be shortened for example using SCC instead of the long term.

-          The figures should increase the resolution and color choices in figure 1 is much better than figures 2 and 3:

o   In figure 1 correct and resize the word Immunodeficiency

o   In Figure 2 the message is not very clear. The legend should be self-explanatory.

o   In figure 3 please check the capital letters.

-          Line 305: change “utmost”

-          Add if any study are available regarding the genetic background/signature that can have a prognostic value in response to immunotherapy (as good/bad responders to the treatment).

-          Some description or introduction about the concept of immunotherapy could be also very useful.

-          In the conclusions section adding some details(numbers) about any successful prevention strategy should be also a good point to consider.

Reviewer 2 Report

Tuba M. Ansary and colleagues present a quality and well-written review manuscript describing immunotherapy for the treatment of squamous cell carcinoma with the focus on potential benefits and challenges.

Authors explain that immunotherapy possesses significant therapeutic benefits for patients with metastatic or locally advanced tumors not eligible for surgery or radiotherapy to avoid the potential toxicity caused by the chemotherapies. Despite the high tolerability and efficiency, some challenges come along, for example, resistance to immunotherapy, less availability of the biomarkers, and appropriate patient selection. 

In this manuscript authors aimed to accumulate the evidence regarding the genetic alterations related to squamous cell carcinoma, the factors that contribute to the potential benefits of immunotherapy, and the challenges to follow this treatment regime.

Authors overview the etiology of squamous cell carcinoma with focus on external and internal factors that are associated with frequent genetic mutations. They also cover available treatments which include factors that determine the potential benefits of immunotherapy, as well as potential challenges.

Finally, authors conclude that due to poor outcome and limited treatment options, squamous cell carcinoma remains a challenging disease for medical professionals. Among other standard therapies, for example, surgical excision, radiotherapy, and chemotherapy, immunotherapies are evolved as a most promising treatment regime as they are highly tolerable and have fewer adverse effects for the squamous cell carcinoma, a highly immunogenic skin cancer. They add that although the number of patients getting benefits out of immunotherapy is still less because of the increased risk of developing resistance and less availability of reliable biomarkers to choose appropriate patients. Authors envisage that advanced studies are needed to overcome these challenges and to propose a combination of treatment modalities so that a higher number of patients can be cured.

Overall, the manuscript is highly valuable for the scientific community and should be accepted for publication after the corrections are made.

==============================

Other comments:

1) Please check for typos throughout the manuscript.

2) Please increase the resolution of the figures (i.e. Fig 3)

3) Fig.1. Last block. “deficiencY”, please correct letter positioning.

4) Section 2.1. With regards to mutations in TP53 gene authors are kindly encouraged to cite the following article that describes various aspects of targeting p53 mutant tumors. DOI: 10.3389/fonc.2020.01460
